**Data Availability Statement:** All relevant data are within the paper and its Supporting Information files.

# Guidance for pediatric use in prescription information for novel medicinal products in the EU and the US

**Helle Christiansen**[1]*, **Marie L. De Bruin**[1,2], **Sven Frokjaer**[1], **Christine E. Hallgreen**[1]

**1** Department of Pharmacy, Faculty of Health and Medical Sciences, Copenhagen Centre for Regulatory Science, University of Copenhagen, Copenhagen, Denmark, **2** Division of Pharmacoepidemiology and Clinical Pharmacology, Utrecht Institute for Pharmaceutical Sciences, Utrecht University, Utrecht, The Netherlands

* hchristiansen@sund.ku.dk

## Abstract

Pediatric legislations in the European Union (EU) and the United States (US) have increased medicines approved for use in the pediatric population. Despite many similarities between these frameworks, the EU Paediatric Regulation more often provides regulators with a mandate to require pediatric drug development for novel medicinal products compared to US regulators. If used, this could give rise to differences in the guidance for pediatric use provided for clinicians in the two regions. However, the level of discordance in the guidance for pediatric use between the two regions is unknown. This cross-sectional study compares guidance for pediatric use in the EU Summary of Product Characteristics (SmPC) and the US Prescription Information (USPI) on the level of indications granted for novel medicinal products approved after the pediatric legislations came in to force in both regions. For all indications granted as of March 2020 for novel medicinal products approved in both regions between 2010 and 2018, we compared the guidance for pediatric use in the EU SmPC and the USPI. The guidance for pediatric use differed for 18% (61/348) of the listed indications covering 21% (45/217) of the products, but without the guidance being contradictory. Where guidance differed, an equal share was observed for indications with a higher level of information for pediatric use in one region over the other (49% (30/61) in the US; 51% (31/61) in the EU). The discrepancies in pediatric information could be explained by differences in regulations for 21% (13/61) of the indications. Only a few conditions and diseases (EU $n = 4$; US $n = 1$) were observed to cover potential pediatric use outside the approved adult indication. Although the EU Paediatric Regulation more often provides regulators a mandate for requiring pediatric drug development as compared to the US PREA, this was not reflected in the prescription information approved by the two regulatory authorities.

**Funding:** HC PhD project was funded by a grant from Lundbeck Pharma A/S to the Copenhagen Centre for Regulatory Science. URL to funder homepage: https://www.lundbeck.com/dk The funders had no role in study design, data collection and analysis, decision to publish, or preparation of the manuscript.

**Competing interests:** The authors have declared that no competing interests exist.

## Introduction

Both the European Union (EU) and the United States (US) have imposed regulations facilitating pediatric drug development to improve the health of children. One direct output of the pediatric legislation in both regions is the inclusion of knowledge generated from agreed pediatric drug development plans into the approved prescribing information as, e.g., either an indication, contraindication, or clinical data [1–5]. The knowledge generated should be included even if the development shows a negative benefit-risk balance for use in the pediatric population.

Only the EU and the US have implemented mandatory pediatric legislation [6] and are considered by some to be the drivers of global pediatric drug development. In the US, Best Pharmaceuticals for Children Act (BPCA) and the Pediatric Research Equity Act (PREA) were introduced in 2002 and 2003, respectively [3, 4]. The PREA makes pediatric drug development mandatory for all new active substances, new indications, new dosage forms, new dosing regimens, or new routes of administration unless a waiver is granted, whereas the BPCA is a voluntary procedure providing incentives of 6-months of additional market exclusivity for the conduct of agreed pediatric studies. In late December 2006, the European Paediatric Regulation (EPR) came into force. It built upon the learnings from the US pediatric legislations and consisted of only one legislation making pediatric drug development mandatory in exchange for a reward of a 6-month supplementary protection certificate (SPC).

While many similarities can be seen between the pediatric legislations in the two regions, differences also exist [7]. A key difference is the scope of the mandatory legislation. In the US, the scope of PREA is restricted to the *proposed indication(s)* for the adult population. In the EU, the Paediatric Committee (PDCO) should use the proposed indication only as a starting point to assess the potential pediatric use of a product *within the condition(s)* of the proposed indication [8]. Also, proposed indications with an orphan drug designation are exempted in the PREA, but not in the EU. Thus, a broader mandatory scope is provided for the requirements of pediatric drug development in the EPR compared to the US mandatory PREA. The Court of Justice in the EU has emphasized that restricting the scope to the proposed adult indication would allow the ignorance of potential pediatric use [9]. In the US, the BPCA can be utilized to request pediatric studies for orphan designated products or potential pediatric use outside the proposed adult indication, however, this is a voluntary procedure.

The EU and the US pediatric legislations have generally been found to be successful in the introduction of products for use in the pediatric population [10–13]. Several studies have found that valuable information was added to the prescription information after the introduction of the pediatric legislations in the EU and the US, which provide clinicians with better information about the safety and efficacy when prescribing products for the pediatric population [10, 14–18]. However, these studies were conducted separately for each regulatory region. Only a few attempts have been made to compare pediatric obligations [19] or the available pediatric prescription information [20] across jurisdictions, this with a focus on pediatric indications. Cross jurisdiction investigations are important because regulatory frameworks and systems should enable and support medicines development in a global context [21] to avoid conflict with other regulatory and societal interests such as inequality of available medicines, especially when populations share similar disease burdens [22].

This study aims to map and compare the guidance for pediatric use in prescription information for novel medicinal products approved by both the Food and Drug Administration (FDA) and EMA after the mandatory pediatric regulations had become effective in both regions. A special focus is given to the description of data from pediatric studies investigating potential pediatric use outside the approved adult indication.

## Methods

### Study cohort

We performed a cross-sectional study on indications granted by March 2020 for all novel medicinal products approved by both EMA and FDA between January 1, 2010, and December 31, 2018. Novel medicinal products (from now on just called "products") were identified using a list of New Active Substances (NAS) maintained for research purposes by the Centre for Innovation in Regulatory Science (CIRS) [23] (see appendix for detailed definition of NAS). All the products were categorized using the WHO Anatomical Therapeutic Chemical (ATC) Classification (5th level, chemical substance) [24] based on recommended international non-proprietary names (rINN), which were used as a starting point to match identical products approved in both regions, followed by subsequent manual quality checks, e.g., in case of multiple potential matches, to assign product pairs for further analysis.

### Data collection

For each identified product pair, we retrieved the most recent US Prescription Information (USPI) and Summary of Product Characteristics (SmPC) together with authorization information, including therapeutic area, date of approval, authorization status, and orphan drug status as of March 2020. The orphan drug designation was included to highlight FDA-approved products exempted from mandatory pediatric regulation in the US. USPIs were retrieved from the **FDA website,** FDA's Center for Drug Evaluation and Research (CDER) [25], or Center for Biologic Evaluation and Research (CBER) [26]. SmPCs were retrieved from the **EMA website,** and additional information was collected from the so-called 'download list' of all European Public Assessment Reports (EPARs) for human and veterinary medicines [27].

For each product pair, SmPCs and USPIs were scrutinized to identify all approved indications (adult and pediatric) in any of the two documents to create the study-unit of product-indication pairs (from now on just called "indications"). Each indication was recorded at the level of condition or disease (depending on the details in the documents) using the Medical Dictionary for Regulatory Activities (MedDRA version 22.1) Preferred Terms (PT) [26]. If several indications had identical conditions or diseases, these were considered as one. For each identified indication, we recorded adult approval status and the type of pediatric information available in the EU and the US, respectively. The type of pediatric information was categorized in a tiered process (Fig 1A) with the first tier looking for pediatric indication or contraindication. If that was not the case, the second tier recorded if there were "recommendations to use" or "recommendation not to use". The third tier recorded if safety and efficacy had not been established, but pediatric clinical data was available, the fourth tier recorded if safety and efficacy had not been established and no pediatric clinical data was available and the fifth tier recorded if no pediatric information was mentioned in SmPC or USPI respectively, as pictured in Fig 1A. The type of pediatric information was captured for each age group of the pediatric population (adolescents (12–18 years), children (2–11 years), infants and toddlers (28 days to 23 months), and term newborn (0–27 days)) as defined by International Conference on Harmonization (ICH) Topic E 11, 2001 [27]. The level of guidance was considered to cover a certain age group if guidance was provided for any age within that group.

If the SmPC or USPI described pediatric information that covered a condition or a disease not included in one of the identified approved indication(s) of the product, the information was recorded in a similar way to approved indications and was further recorded as ´outside the approved adult indication(s)´ to disclose a potential pediatric use not driven by the adult population.

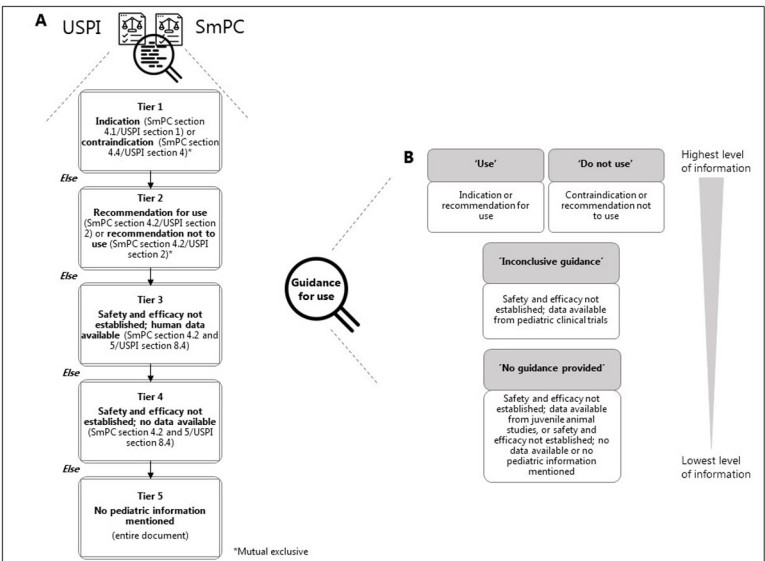

**Fig 1. Categorization and hierarchical grouping of pediatric information.** A) Tiered categorization of pediatric information in the SmPC and the USPI and B) hierarchical grouping of the categorized pediatric information to provide guidance for pediatric use.

The overall guidance for pediatric use provided for condition(s) or disease(s) in the SmPC and USPI was described by hierarchical grouping dividing the type of pediatric information into four categories, as depicted in Fig 1B. The recommendation to 'use' or 'do not use' was considered as the highest level of information; the lowest level of information was 'no guidance provided'. Whenever the categories of guidance for pediatric use differed between the USPI and the SmPC for one or more subsets of the pediatric population, it was defined as a discrepancy in the guidance for pediatric use between the EU and the US. For each discrepancy, it was decided whether USPI or SmPC provided the most information. Also, for each discrepancy, the mandatory pediatric requirements at initial approval of an indication (initial or line extension) were investigated. For this, we used US letters retrieved from the **FDA website** and the PDCO decision mentioned within the EPAR retrieved from the **EMA website**. If a difference existed in the mandatory pediatric requirements (PREA legislation against the EPR) at that point, the mandatory pediatric framework was recorded to be a possible cause of differences seen.

The data collection on authorization information was done by one author (HC). HC first collected data on indication(s) and the level of pediatric information from SmPCs and USPIs, which subsequently was systematically reviewed by another author (CEH). Any disagreement was resolved by consensus.

## Analysis

On the product level, descriptive analyses were performed on orphan designation, product age (approval date–date of data collection), therapeutic area, number of approved indications (adult and pediatric), for each region respectively. A direct comparison of the overall guidance for pediatric use between the US and the EU was performed on indication level. Statistical Mann–Whitney U testing was used to test if the guidance for pediatric use was identical between SmPC and USPI for each age-subgroup of the pediatric population. The significance level was set a priori to a p-value of 0.05. Analyses were performed using the statistical software R, version 3.6.0 (2019-04-26) [28].

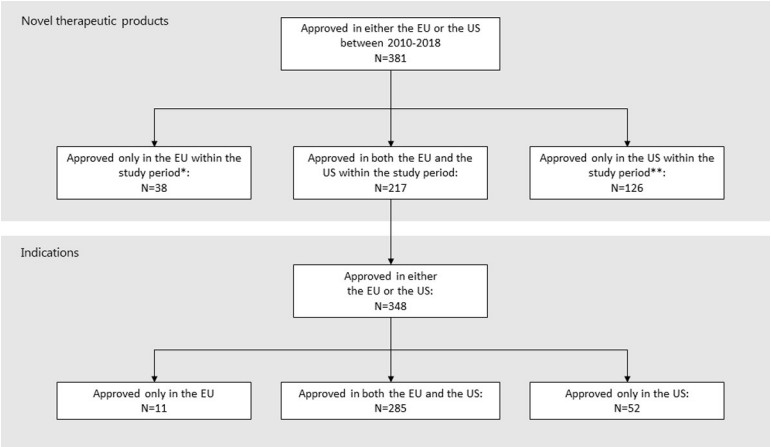

**Fig 2. Selection of cohort.** Flowchart of selection of the new active substances (products) included in the study and the indications for analysis.

## Results

### Characteristics of products

From 1 January 2010 to 31 December 2018, 255 novel products were approved in the EU through the centralized procedure, and 343 novel products were approved in the US. Of these, 217 were approved in both regions (Fig 2). Fifty products had been granted an orphan drug designation for at least one indication in both regions at initial approval. An additional 45 products had been granted an orphan drug designation only in the US (Table 1).

Most products (82%) were approved first in the US. Still, most products (77%) were approved in both regions within one year of each other (for more information see supplementing material). On average, the US product age was higher (5.3 years, IQR: 3.7 years) than the EU products age (4.8 years, IQR: 3.8 years), directly related to the earlier approval in the US.

We found the largest number of products approved within antineoplastic and immunomodulating agents (n = 83, 38%), alimentary tract and metabolism (n = 29, 13%), anti-infective agents (n = 29, 13%), and blood and blood-forming organs (n = 19, 9%) (S1 Table). Differences in pediatric indications between the EU and the US were only seen in six therapeutic areas and more often with more pediatric indications in the US (Table 2). The biggest difference was observed for cancer products. However, this was caused by very few products. The same

**Table 1. Descriptive characteristics at the time of data collection (March 2020) for products approved both in the EU and the US between 2010 and 2018 (*n* = 217).**

|  | EU | | US | |
|---|---|---|---|---|
|  | no. | (%) | no. | (%) |
| Number of products | 217 | (100%) | 217 | (100%) |
| Number of products with orphan drug designation[a] | 50[b] | (23%) | 95[b] | (44%) |
| Median product age (IQR) | 4.8 | (IQR: 3.8) | 5.3 | (IQR: 3.7) |
| First region of approval | 39 | (18%) | 178 | (82%) |
| Number of indications (adult and/or pediatric) per product (median) | 1 | (range 1–7) | 1 | (range 1–16) |

[a]At initial approval
[b]Of which 50 in both regions.

**Table 2. Descriptive overview of therapeutic areas of products where differences exist in the pediatric indications approved in the EU and the US.** Ordered after the therapeutic area for which most of the total active products were approved. See the S1 Table for an overview of the therapeutic areas of all products.

| Anatomical main group | Number of products in the ATC group | Products with a pediatric indication(s) | | | | Products with a pediatric indication (s) in both regions | |
|---|---|---|---|---|---|---|---|
| | | EU | | US | | | |
| | no. | no. | (%)[a] | no. | (%)[a] | no. | (%)[a] |
| Antineoplastic and immunomodulating agents | 83 | 7 | (8%) | 11 | (13%) | 6 | (7%) |
| Anti-infective for systemic use | 29 | 10 | (34%) | 13 | (45%) | 10 | (34%) |
| Alimentary tract and metabolism | 29 | 11 | (38%) | 10 | (34%) | 10 | (34%) |
| Nervous system | 14 | 2 | (14%) | 3 | (21%) | 2 | (14%) |
| Cardiovascular system | 7 | 1 | (14%) | 2 | (29%) | 1 | (14%) |
| Sensory organs | 4 | 1 | (25%) | 2 | (50%) | 1 | (25%) |

[a]calculated from the total products approved within a therapeutic area.

pattern was seen for products with a pediatric indication, albeit with a more prominent role for agents for diseases in blood and blood-forming organs.

## Pediatric indications and guidance for use

For the 217 products included in this study, we identified 348 indications approved either initial or after the initial approval; of these 285 were granted in both regions, 11 only in the EU and 52 only in the US (Fig 2). The median number of indications granted per product in both regions was one (US: range 1–16, EU: range 1–7, see Table 1).

A pediatric indication was granted for at least one pediatric age group for 68 indications granted for 59 products approved in the US (Table 3). Of these, 54 indications were also

**Table 3. Descriptive overview of the approved pediatric indications and pediatric guidance for use by products and indications in the EU and the US.**

| | EU | | | | US | | | |
|---|---|---|---|---|---|---|---|---|
| | Products | | Indications | | Products | | Indications | |
| | no. | (%) | no. | (%) | no. | (%) | no. | (%) |
| Total | 217 | (100%) | 296 | (100%) | 217 | (100%) | 337 | (100%) |
| Only approved for the pediatric population | 2 | (1%) | 2 | (1%) | 2 | (1%) | 2 | (1%) |
| Adult and pediatric indication[a] for at least one pediatric subgroup: | 50 | (23%) | 54 | (18%) | 59 | (27%) | 68 | (20%) |
| - Adolescents(12–18 years[b])[c] | 50 | (100%) | 54 | (100%) | 59 | (100%) | 68 | (100%) |
| - Children (2–11 years) [c] | 33 | (66%) | 34 | (63%) | 45 | (76%) | 50 | (75%) |
| - Toddlers and infants (27 days-23 months) [c] | 22 | (44%) | 23 | (43%) | 26 | (44%) | 30 | (54%) |
| - Term newborn (0–26 days)[c] | 17 | (34%) | 18 | (33%) | 16 | (27%) | 19 | (30%) |
| Only approved for the adult population | 165[d] | (76%) | 240 | (81%) | 156[d] | (72%) | 267 | (66%) |
| - Recommendation *not* to use | 8 | (5%) | 15 | (6%) | 5[e] | (3%) | 10[e] | (4%) |
| - Pediatric data available from clinical trial(s) for one or more subgroup | 15 | (9%) | 18 | (8%) | 4 | (3%) | 6 | (2%) |
| - No guidance for pediatric use was provided[f] for any pediatric subgroup | 142 | (86%) | 207 | (86%) | 147 | (94%) | 251 | (94%) |

[a]For products, at least one pediatric indication

[b]12-17 years in the US

[c]Pediatric indication can cover several age group, numbers do not add up to 100%

[d]Only products without a pediatric indication

[e]Two indications had a contraindication

[f]Safety and efficacy had not been established, no human data was available for the pediatric population.

**Table 4. Level of guidance for pediatric use for indications in the EU and the US per age group (percentages calculated from the total number of indications, _n_ = 348).**

| | Use | | | | Do not use | | | | Human data available | | | | No guidance provided | | | | P-value[a] | Discrepancies | |
|---|---|---|---|---|---|---|---|---|---|---|---|---|---|---|---|---|---|---|---|
| | EU | | US | | EU | | US | | EU | | US | | EU | | US | | | | |
| | no. | (%) | no. | % | no. | (%) | no. | (%) | no. | (%) | no. | (%) | no. | (%) | no. | (%) | | no. | (%) |
| Adolescents | 56 | (16%) | 70 | (20%) | 12 | (3%) | 5 | (1%) | 21 | (6%) | 6 | (2%) | 259 | (75%) | 267 | (77%) | 0.61 | 40 | (11%) |
| Children | 36 | (10%) | 52 | (15%) | 15 | (4%) | 10 | (3%) | 17 | (5%) | 4 | (1%) | 280 | (81%) | 282 | (82%) | 0.98 | 35 | (10%) |
| Infants and toddlers | 25 | (7%) | 32 | (9%) | 17 | (5%) | 10 | (3%) | 11 | (3%) | 4 | (1%) | 295 | (85%) | 302 | (87%) | 0.39 | 26 | (7%) |
| Term newborns | 19 | (5%) | 21 | (6%) | 17 | (5%) | 10 | (3%) | 9 | (3%) | 4 | (1%) | 303 | (88%) | 313 | (90%) | 0.15 | 26 | (7%) |
| Any ped. age | | | | | | | | | | | | | | | | | | 61[b] | (18%) |

[a]Mann-Whitney U test for discrepancies within each age-group

[b]multiple age groups per indication possible, numbers in the column do not **add up to 61.**

granted in at least one pediatric age group in the EU covering 50 products. In both regions, all pediatric indications included adolescents. Pediatric indications were less available in younger age groups such as toddlers, infants, and term newborns. Two indications approved for the active substances cerliponase alfa (Brineura) and dinutuximab (Unituxin) were approved in both regions solely for pediatric use. The absolute majority of the indications (EU 240; US 267) were approved only for the adult population; mostly without guidance for pediatric use. In the US, two indications (with the active substance linaclotide) were not recommended for use in the pediatric population based on contraindications. In the EU, use in children and adolescents was explicitly not recommended for this product. No pediatric contraindications were seen in the EU.

## Discrepancies in guidance for pediatric use

Overall, we found that guidance for pediatric use in the SmPC and USPI differed in 18% (n = 61/348) of the indications representing 21% (n = 45/217) of the novel products (Table 4). However, these differences were not statistically significant. Most discrepancies were found for adolescents (n = 40, 11%) and children (n = 35, 10%). In the subset of indications without an orphan drug designation in the US and/or a pediatric indication outside of the adult indication, the results showed the same pattern (see S5 Table). In general, no contradictory guidance was observed (e.g., guidance for 'use' in one region as compared to guidance of 'do not use' in the other). With adolescents as an example (Table 5), most often cases of discrepancy had guidance for 'use' in USPI (n = 18) or 'do not use' in SmPC (n = 10) in combination with 'safety and efficacy not established' in the opposite document (Table 5, see Box 1 for examples). The same pattern, as described above, was observed in the younger age groups of the pediatric population (see supplementary material). In addition, the discrepancies had an equal share

**Table 5. Concordance of the level of guidance for pediatric use in the SmPC and the USPI for adolescents (_n_ = 348).** For the other age groups, please see S2–S4 Tables.

| | USPI Use | Do not use | Human data available | No guidance provided | Total |
|---|---|---|---|---|---|
| **SmPC** | | | | | |
| Use | 52 | 0 | 0 | 4 | 56 |
| Do not use | 0 | 3 | 0 | 9 | 12 |
| Human data available | 7 | 1 | 6 | 7 | 21 |
| No guidance provided | 11 | 1 | 0 | 247 | 259 |
| Total | 70 | 5 | 6 | 267 | 348 |

## Box 1. Examples of discrepancies in guidance for pediatric use between the SmPC and the USPI

Ixekizumab (Taltz) was indicated for plaque psoriasis in the adult population in both EU and US. In US, it was also indicated for pediatric patients 6 years of age and older, whereas in the EU it was stated that safety and efficacy had not been established in the pediatric population.

Selexipag (Uptravi) was indicated for pulmonary arterial hypertension (PAH) in the adult population in both the EU and US for which safety and efficacy had not established in the pediatric populations. However, based on results from animal studies indicating an increased risk of intussusception with unknown clinical relevance, pediatric use was not recommended in the EU.

For descriptions of all cases of discrepancy, see Tables 5 and 6.

between USPIs and SmPCs providing a higher level of information for pediatric use in one region compared to the other (49% (n = 30) in the US; 51% (n = 31) in EU, see Tables 6 and 7). In only 21% (13/61) of the discrepancies, the mandatory pediatric regulations offered a possible explanation (Tables 6 and 7).

### Pediatric data available for conditions and diseases outside approved adult indication

A summary of pediatric data available for condition(s) and disease(s) outside of the approved adult indication is provided in Table 8. The SmPC of four products with the active substances cabazitaxel, pembrolizumab, sonidegib, and lurasidone, described data from pediatric studies investigating one or more diseases not covered by the approved adult indications. Of these, cabazitaxel also described pediatric data for disease outside the approved adult indications in USPI.

## Discussion

Even though the legal framework of mandatory pediatric legislation in the EU and the US provide a basis for differences in pediatric research obligations for the same product in the two regions [7], we observed only a few discrepancies between the guidance for pediatric use in the prescription information approved in the respective regions. Only 18% of the indications representing 21% of novel products approved in both the EU and the US between 2010 and 2018 differed in the guidance for pediatric use. Furthermore, an equal share was seen between discrepancies having a higher level of information for pediatric use in one region compared to the other (49% (n = 30) in the US; 51% (n = 31) in the EU). Also, this study showed that potential pediatric use outside an adult indication was rarely covered in the SmPC and the USPI.

The low number of discrepancies in guidance for pediatric use between SmPCs and USPIs could be explained if the broader mandate for mandatory pediatric requirements in the EPR compared to PREA is not used in practice. However, almost half of the products in our study had an orphan drug designation in the US, exempting them from the requirements of mandatory pediatric drug development through US PREA. Nonetheless, a comparison of guidance for pediatric use between SmPCs and USPIs in this subset showed the same pattern as for the

**Table 6. Summary of discrepancies in the guidance for pediatric use for indications where USPI provides more guidance than SmPC[a] (*n* = 30).** Ordered by guidance for pediatric use in the US and alphabetically by active substances name.

| Product (rINN) | ATC code | Summary of indication[b] | Summary of differences | Pediatric subgroup for which the difference exists | Can differences be explained by differences in the mandatory pediatric requirements at initial approval of indication? |
|---|---|---|---|---|---|
| Recommendation to use in USPI[c] | | | | | |
| Avelumab | L01XC31 | Merkel cell carcinoma (MCC) | Indicated for use in adults and adolescents in the US, but only for adults in the EU. | Adolescents | No |
| Benralizumab | R03DX10 | Eosinophilic asthma | Indicated for use in adults and adolescents in the US, but only for adults in the EU. Data are available for adolescents in the EU. | Adolescents | No |
| Bictegravir, emtricitabine, tenofovir alafenamide, fumarate | J05AR20 | Hiv-1 | Indicated for use in adults, adolescents, and children in the US, but only for adults in the EU. | Adolescents and children | No |
| Blinatumomab | L01XC19 | Acute lymphoblastic leukemia | Indicated for use in adults and the entire pediatric population in the US. In the EU, term newborns are not covered by an indication. | Term newborn | No |
| Ceftazidime / avibactam | J01DD52 | Complicated intra-abdominal infections | Indicated for use in adults, adolescents, children, toddlers, and infants in the US, but only for adults in the EU. | Adolescents, children, toddlers, and infants | No |
| | | Complicated urinary tract infection | Data were available for pediatric patients in the EU. | | |
| Dinutuximab | L01XC16 | Neuroblastoma | Indicated for use in the entire pediatric population in the US. In the EU, term newborns are not covered by an indication. | Term newborn | No |
| Ipilimumab | L01XC11 | Metastatic Colorectal Cancer | Indicated for use in adults and adolescents in the US. The indication was not approved in the EU and no pediatric information was mentioned. | All subgroups | No |
| Ixekizumab | L04AC13 | Plaque psoriasis | Indicated for use in adults, adolescents, and children in the US, but only for adults in the EU. | Adolescents and children | No |
| Lurasidone | N05AE05 | Bipolar I disorder | Indicated for use in adults, adolescents, and children in the US, but only for adults in the EU. | Adolescents and children | No |
| | | | Data were available for pediatric patients in the EU. | | |
| | | Schizophrenia | Indicated for use in adults and adolescents in the US, but only for adults in the EU. | Adolescents | No |
| | | | Data were available for pediatric patients in the EU. | | |
| Metreleptin | A16AA07 | Congenital or acquired generalized lipodystrophy. | Indicated for use in adults and the entire pediatric population in the US. In the EU, toddlers and infants, and term newborns are not covered by an indication. | Toddler and infants and term newborn | No |
| | | | Data were available for pediatric patients in the EU. | | |
| Nivolumab | L01XC17 | Metastatic colorectal cancer | Indicated for use in adults and adolescents in the US. The indication was not approved in the EU and no pediatric information was mentioned. | Adolescents | No |
| Nonacog beta pegol | B02BD04 | Bleeding, hemophilia B | Indication for use in adults and the entire pediatric population in the US, but only for adults and adolescents in the EU. | Children, toddlers, infants, and term newborn | No |
| | | | Data were available for pediatric patients in the EU. | | |

(*Continued*)

**Table 6.** (Continued)

| Product (rINN) | ATC code | Summary of indication[b] | Summary of differences | Pediatric subgroup for which the difference exists | Can differences be explained by differences in the mandatory pediatric requirements at initial approval of indication? |
|---|---|---|---|---|---|
| Pembrolizumab | L01XC18 | Primary mediastinal large b-cell lymphoma | Indicated for use in adults and the entire pediatric population in the US. | All subgroups | No |
| | | Microsatellite instability-high cancer | Indication not approved in EU. Data were available for pediatric patients in the EU. | | |
| | | Merkel cell carcinoma | Indicated for use in adults and the entire pediatric population in the US. | All subgroups | No |
| | | | The indication was not approved in the EU and no pediatric information was mentioned. | | |
| | | Classical Hodgkin lymphoma | Indicated for use in adults and the entire pediatric population in the US, but only for adults in the EU. | All subgroups | No |
| | | | Data were available for pediatric patients in the EU. | | |
| Perampanel | N03AX22 | Epilepsy | Indicated for use in adults, adolescents, and children in the US, but only for adults and adolescents in the EU. | Children | No |
| Recombinant human nerve growth factor (rhngf) | S01XA24 | Neurotrophic keratopathy | Indicated for use in adults, adolescents, and children in the US, but only for adults in the EU. | Adolescents and children | No |
| Ruxolitinib | L01XE18 | Acute graft versus host disease | Indicated for use in adults and adolescents in the US. The indication was not approved in the EU and no pediatric information was mentioned. | Adolescents | No |
| Sacubitril / valsartan | C09DX04 | Chronic heart failure | Indicated for use in adults, adolescents, children, toddlers, and infants in the US, but only for adults in the EU. | Adolescents, children, toddlers, and infants | No |
| Sofosbuvir | J05AX15 | Chronic hepatitis c | Indicated for use in adults, adolescents, and children in the US, but only for adults and adolescents in the EU. | Children | No |
| Sofosbuvir / ledipasvir | J05AX65 | Chronic hepatitis c | Indicated for use in adults, adolescents, and children in the US, but only for adults and adolescents in the EU. | Children | No |
| Sofosbuvir / velpatasvir | J05AP55 | Chronic hepatitis c virus | Indicated for use in adults, adolescents, and children in the US, but only for adults in the EU. | Adolescents and children | No |
| Tezacaftor, ivacaftor | R07AX31 | Cystic fibrosis | Indicated for use in adults, adolescents, and children in the US, but only for adults and adolescents in the EU. | Children | No |
| Voretigene neparvovec | S01XA27 | Retinal dystrophy | Indicated for use in adults and the entire pediatric population in the US. In the EU, toddlers and infants, and term newborns are not covered by an indication. | Toddler and infants and term newborn | No |
| Recommendation *not* to use in USPI | | | | | |
| Denosumab | M05BX04 | Treatment-induced bone loss in Women with breast cancer | No recommendation for use in the entire pediatric population in the US. The indication was not approved in the EU and no pediatric information was mentioned. | All subgroups | No |
| | | | For the other indications in the EU, the pediatric guidance was equal to that in the US; no use was recommended. | | |

(*Continued*)

**Table 6.** (Continued)

| Product (rINN) | ATC code | Summary of indication[b] | Summary of differences | Pediatric subgroup for which the difference exists | Can differences be explained by differences in the mandatory pediatric requirements at initial approval of indication? |
|---|---|---|---|---|---|
| Dulaglutide | A10BJ05 | Type II diabetes mellitus | No recommendation for use in the entire pediatric population in the US. In the EU, no safety and efficacy have been established. | All subgroups | No |
| Ocriplasmin | S01XA22 | Vitreomacular adhesion | No recommendation for use in the entire pediatric population in the US based on a single clinical trial. In the EU, no guidance was provided but available data from a clinical trial was described. | All subgroups | Possible. A PSP agreed for the entire pediatric population in the US, whereas a full waiver was granted in the EU. |

[a]More guidance defined as the guidance for pediatric use in USPI was "use" or "do not use" compared to "human data available" or "safety and efficacy not established" in SmPC.

[b]Summary only covers the indication for which a difference exists. Several indications can have been granted for the active substance.

[c]All based on a pediatric indication for at least one subgroup of the pediatric population.

entire study cohort. Future studies should investigate if products exempted from the US PREA received a waiver in the EU, which could explain the low number of discrepancies in guidance for pediatric use between the SmPC and the USPI; Or if pediatric requirements were posted by the EMA regulators, this led to a request for voluntary pediatric drug development through BPCA in the US (spillover effect). However, one study found that of the 40 Written Request issued through US BPCA for oncology products since 2001, only three products have been approved for use in the pediatric population [29] suggesting that the US BPCA only provide a small contribution to the aligned guidance for pediatric use for oncology products found in our study.

In August 2007, a pediatric cluster was established between the EMA and FDA. This initiative aimed to facilitate harmonization in the regulatory requirements and the conduct of pediatric clinical studies in both regions [30, 31]. This cluster may explain the low level of discrepancies in guidance for pediatric use seen between the SmPC and the USPIs. One study, exanimating the EMA Paediatric Committee (PDCO) decisions and FDA Pediatric Review Committee (PeRC) recommendations, showed a high similarity (86%) in a subset (n = 80, 20%) of waiver applications submitted to EMA from January 2007 through December 2013 [19].

In the majority (48/61) of the discrepancies in the guidance for pediatric use, we did not observe these to be a result of the differences in the mandatory requirements for pediatric development in the US and the EU. Rather, discrepancies in guidance for pediatric use seemed to be caused by differences in regulatory decisions or company strategy. As an example of a regulatory decision, nonacog beta pegol (EU: Refixia, US: Rebinyn) was indicated for treatment and prophylaxis of bleeding in adults and the entire pediatric population with hemophilia B in the US, but in the EU, the indication was restricted to adults and adolescents. However, the pediatric data available in the SmPC were similar to the pediatric data presented in the USPI. Hence, this difference seems to be caused by different interpretations of data by the regulatory agencies. The difference in agency conclusion on efficacy has previously been found to be the most common reason for initial discordance in MA decision outcome between the two regions [32]. We also observed discrepancies to be caused by differences in the pediatric data available for review by regulatory agencies; this root is the second most common source of divergent FDA and EMA outcomes [32]. Several discrepancies could be assigned to a

**Table 7. Summary of discrepancies in the guidance for pediatric use for indications where SmPC provides more guidance than USPI[a] (*n* = 31).** Ordered by the guidance for pediatric use in the EU in which the difference exists and alphabetically by active substances name.

| Product (rINN) | ATC code | Indication summary[b] | Summary of differences[b] | Pediatric subgroup for which the difference exists | Can differences be explained by differences in the mandatory pediatric requirements at initial approval of indication? |
|---|---|---|---|---|---|
| Recommendation to use in SmPC[c] | | | | | |
| Ceftaroline fosamil | J01DI02 | Community-acquired pneumonia | Indicated for use in adults and the entire pediatric population in the EU. In the US, term newborns are not covered by the indication. | Term newborn | No |
| Elosulfase alfa (Recombinant human n-acetylgalactosamine-6-sulfatase (rhgalns)) | A16AB12 | Mucopolysaccharidosis iv | Indicated for use in the entire pediatric population in the EU. In the US, toddlers and infants, and term newborns are not covered by the indication. | Toddler and infants and term newborn | Possible. Pediatric drug development is required for the entire pediatric population in the EU, whereas ODD was granted in the US. |
| Fidaxomicin | A07AA12 | C. difficile diarrhea | Indicated for use in adults and the entire pediatric population in the EU. In the US, term newborns are not covered by the indication. | Term newborn | Possible. Pediatric drug development is required for the entire pediatric population in the EU, but only for adolescents and children in the US. |
| Fluticasone furoate / vilanterol | R03AK10 | Asthma | Indicated for use in adults and adolescents in EU. The indication was not approved in the US and no pediatric information was mentioned. | All subgroups | No. |
| Metreleptin | A16AA07 | Familial or acquired partial lipodystrophy (LD) | Indicated for use in adults and adolescents in the EU. The indication was not approved in the US and no pediatric information was mentioned. | All subgroups | No |
| Migalastat hydrochloride | A16AX14 | Alpha galactosidase a deficiency | Indicated for use in adults and adolescents aged 16 years in the EU, but only for adults in the US. | Adolescents | Possible. Pediatric drug development is required for adolescents and children in the EU, whereas ODD was granted in the US. |
| Sebelipase alfa | A16AB14 | Enzyme replacement therapy | Indicated for use in adults and the entire pediatric population in the EU. In the US, term newborns are not covered by the indication. | Term newborn | Possible. Pediatric drug development is required for the entire pediatric population in the EU, whereas ODD was granted in the US. |
| Vandetanib | L01XE12 | Medullary thyroid cancer | Indicated for use in adults, adolescents, and children in the EU, but only for adults in the US. | Adolescents and children | Possible. Pediatric drug development for adolescents in the EU, whereas ODD in the US. |
| Velaglucerase alfa | A16AB10 | Enzyme replacement therapy | Indicated for use in adults and the entire pediatric population in the EU. In the US, toddlers and infants, and term newborns are not covered by the indication. | Toddler and infants and term newborn | Possible. Pediatric drug development for adolescents and children in the EU, whereas ODD in the US. |
| Recommendation *not* to use in SmPC | | | | | |
| Denosumab | M05BX04 | Osteoporosis (postmenopausal and steroid-induced)[d] Bone loss | Not recommended for use in the entire pediatric population in the EU. In the US, adolescents were not covered by the recommendation *not* to use. | Adolescents | No |

(*Continued*)

**Table 7.** (Continued)

| Product (rINN) | ATC code | Indication summary[b] | Summary of differences[b] | Pediatric subgroup for which the difference exists | Can differences be explained by differences in the mandatory pediatric requirements at initial approval of indication? |
|---|---|---|---|---|---|
| Eravacycline | J01AA13 | Complicated intra-abdominal infection | Indicated for only adults in both EU and US. | Children, toddlers and infants, and term newborn | No |
| | | | Not recommended for use in children younger than the age of 8 years in the EU. In the US, no guidance was provided. | | |
| Ipilimumab | L01XC11 | Advanced renal cell carcinoma | Indicated for only adults in both EU and US. Not recommended for use in pediatric patients below 12 years in the EU. In the US, no guidance was provided. | Children, toddlers and infants, and term newborn | No |
| | | Advanced melanoma and Advanced renal cell carcinoma | | | |
| Lenvatinib | L01XE29 | Hepatocellular carcinoma | Indicated for use in adults and adolescents in both EU and US. Not recommended for use in pediatric patients below the age of 2 years in the EU. In the US, no guidance was provided. | Toddlers and infants, term newborns. | No |
| | | Thyroid cancer | | | |
| Midostaurin | L01XE39 | Acute myeloid leukemia | Only indicated for adults in both EU and US. Not recommended for use in the entire pediatric population in the EU. In the US, no guidance was provided. | All subgroups | Possible. |
| | | | | All | Pediatric drug development is required for adolescents and children in the EU, whereas ODD was granted in the US. |
| | | Systemic mastocytosis | | | No |
| Riociguat | C02KX05 | Chronic thromboembolic pulmonary hypertension | Indicated for only adults in both EU and US. Not recommended for use in the entire pediatric population in the EU. In the US, no guidance was provided. | All subgroups | Possible. |
| | | Pulmonary arterial hypertension | | | Pediatric drug development is required for the entire pediatric population in the EU, whereas ODD was granted in the US. |
| Selexipag | B01AC27 | Pulmonary arterial hypertension | Indicated for only adults in both EU and US. Not recommended for use in the entire pediatric population in the EU. In the US, no guidance was provided. | All subgroups | Possible. |
| | | | | | Pediatric drug development is required for the entire pediatric population in the EU, whereas ODD was granted in the US. |
| Data available in SmPC | | | | | |
| Brentuximab vedotin | L01XC12 | Classical Hodgkin lymphoma | Data were available in SmPC from a pediatric study (36 patients aged 7–17 years). Data were not available in USPI. | Adolescent and children | Possible. |
| | | Anaplastic large cell lymphoma | | | Pediatric drug development is required for adolescents and children in the EU, whereas ODD was granted in the US. |
| Brivaracetam | N03AX23 | Epilepsy, partial seizures | Indicated for use in adults, adolescents, and children in both EU and US. | Toddlers and infants | No |
| | | | Data were available in SmPC for toddlers and infants. Data were not available in USPI. | | |
| Canagliflozin | A10BK02 | Type II diabetes mellitus | Data were available in SmPC from a pediatric study (patients aged 10–18 years). Data were not available in USPI. | Adolescent and children | No |

(*Continued*)

**Table 7.** (Continued)

| Product (rINN) | ATC code | Indication summary[b] | Summary of differences[b] | Pediatric subgroup for which the difference exists | Can differences be explained by differences in the mandatory pediatric requirements at initial approval of indication? |
|---|---|---|---|---|---|
| Empagliflozin | A10BK03 | Type II diabetes mellitus | Data were available in SmPC from a pediatric study (patients aged 10–18 years). Data were not available in USPI. | Adolescent and children | No |
| Ezogabine (Retigabine) | N03AX21 | Epilepsy, partial seizures | Data were available in SmPC from a pediatric study (5 patients aged 12–18 years). Data were not available in USPI. | | No |
| Linagliptin | A10BH05 | Type II diabetes mellitus | Data were available in SmPC from a pediatric study (patients aged 10–18 years). Data were not available in USPI. | Adolescent and children | No |
| Tedizolid phosphate | J01XX11 | Absssi | Data were available in SmPC from a pediatric study (20 patients aged 12–17 years). Data were not available in USPI. | Adolescent | No |

[a]More guidance defined as the guidance for pediatric use in SmPC was "use", "do not use" or "data available" compared to "no guidance for use" in the USPI.

[b]Summary only covers the indication for which a difference exists. Several indications can have been granted for the active substance.

[c]All based on a pediatric indication for at least one subgroup of the pediatric population

[d]Three indications gathered on the condition (osteoporosis), numbers of indications in the table do not add up to 31 indications.

missing indication in one region, which could be a result of marketing strategies from a company not seeking an indication.

The clinical consequences of the differences in guidance for pediatric use are difficult to address. Especially since the guidance for pediatric use in USPIs and SmPCs might not include all knowledge available to the treating physician. At the initial and supplemental approval, knowledge from ongoing studies and studies conducted in other settings for which study results have not been published (e.g., with a non-commercial sponsor) is not included. In such studies, important guidance for pediatric use could be discovered and included in, e.g., treatment guidelines. Also, a small survey conducted in the US, observed that many pediatricians were unaware of pediatric label changes [33]. In addition, based on the lack of information available to doctors, a long tradition of off-label prescription exist in the treatment of the pediatric population [34]. Such prescription might reduce the potential clinical consequences of the differences in guidance for pediatric use between SmPCs and USPIs found in our study.

The finding of only a few cases that meet potential unmet pediatric needs outside an adult indication was consistent with the results of earlier studies showing adult indications to be the drivers of pediatric drug development in the EU [10, 18, 35–37] and the US [38, 39].

A post hoc examination of the four cases where pediatric data was described for conditions and diseases outside an adult indication suggested that three cases (cabazitaxel, sonidegib, and lurasidone) were likely to be a result of the pediatric legislations as obligations had been posted by either the US BPCA (cabazitaxel) or PREA (lurasidone) or the EPR (sonidegib). However, the USPI for pembrolizumab (Keytruda) described four pediatric indications whereas the SmPC only contained information on the doses administrated and the preliminary safety profile resulting from an ongoing clinical trial; however, no requirements for a pediatric drug

**Table 8. Summary of pediatric data available for a condition and disease outside the adult indications in the SmPC (n = 4) and the USPI (n = 1).** Pediatric-only indications not included.

| Product (rINN) | ATC code | Condition/disease of pediatric investigation | Adult indication(s) approved for product | Pediatric information | |
|---|---|---|---|---|---|
| | | | | USPI | SmPC |
| **Both: Condition/disease of pediatric investigation outside the adult indication in both EU and US** | | | | | |
| Cabazitaxel | L01CD04 | High-grade glioma (HGG) or diffuse intrinsic pontine glioma (DIPG). | Castration-resistant prostate cancer | No guidance for pediatric use was provided. | No guidance for pediatric use was provided. |
| | | | | Data available from 39 pediatric patients (ages 3 to 18 years) receiving cabazitaxel. | Data available from an open-label, multi-center, phase 1/2 study conducted in a total of 39 pediatric patients (aged between 4 to18 years for the phase 1 part of the study and between 3 to 16 years for the phase 2 part of the study). |
| **EU only: Condition/disease of pediatric investigation outside the adult indication in EU** | | | | | |
| Pembrolizumab | L01XC18 | Various solid tumors (e.g., advanced melanoma, glioblastoma multiforme, neuroblastoma, and osteosarcoma) | In the US, it is indicated for many different cancer treatments only in the adult population (10 diseases)[a] but also in both the adult and the pediatric population (4 diseases)[b]. | The condition/disease was covered by indications in both adult and pediatric patients. | No guidance for pediatric use was provided. |
| | | | In the EU, it is indicated for different cancer treatments only in the adult population (10 diseases)[c]. | | Data were available from a phase I/II study conducted in a total of 154 pediatric patients (60 children aged 6 months to less than 12 years and 94 adolescents aged 12 years to 18 years). |
| Sonidegib | L01XX48 | Medulloblastoma or other tumors potentially dependent on the Hedgehog (Hh) signaling pathway | Basal cell carcinoma | The condition/disease was not covered by indications in both the adult and the pediatric patients, and no data was mentioned | No guidance for pediatric use was provided. |
| | | | | | Data were available from two clinical studies (phase I/II and phase II) involving a total of 62 pediatric patients. |
| Lurasidone | N05AE05 | Major depressive episode associated with bipolar I disorder | Schizophrenia in adults, adolescents, and children in the US, but only adults in the EU. Major depressive episode associated with bipolar I disorder is only approved in the US. | Indicated for use in adult and pediatric patients (10 to 17 years). | No guidance for pediatric use was provided. |
| | | | | | Data were available from a 6-week multicentre, randomized, double-blind, placebo-controlled, clinical trials in children and adolescent patients (10–17 years of age). |

development were posted by either the EMA or the FDA. This is in alignment with studies showing evidence of waived pediatric drug development for relevant pediatric targets [35, 36].

The results should be interpreted within the limitations of this study. The study is a snapshot of time where products were followed on average for approximately five years since the initial approval. A median time-lag of 4–5 years between the initial approval and the first supplemental pediatric indication has been shown in both the EU and the US [20]. If followed for a longer time, more products having guidance for pediatric use would be expected. Second, the effects of pediatric requirements resulting from supplemental adult indications may contribute to an underestimation of the amount of pediatric indication because these would have an even shorter follow-up time than the calculated total follow-up. However, the absolute majority (73%, n = 157) of the products in the study cohort only have a single indication. Third, indications with the same condition and disease were collapsed, and the number of indications in USPIs and SmPCs is higher than the recorded number in this study. Last, categorization of the guidance for pediatric use could have looked different if, e.g., 'no relevant use' was interpreted as a statement of ´do not use´ instead of ´safety and efficacy not established´. A strength of the study is the inclusion of the entire population of novel products approved since both the EU and the US pediatric legislations came into effect.

Further research is needed to assess the extent to which pediatric development has resulted in new posology guidance for the pediatric and possible new age-appropriate formulations. Also, follow-up research is needed to study the concordance of obligations for pediatric drug development in the EMA and the FDA. This process could shed more light on whether one region is a driver of the regulatory pediatric drug development, or if different requirements result in inconclusive guidance for use (e.g., human data available) in both regions and perhaps not provide any added value to the health of the pediatric patients. Recently, the US PREA was amended to allow requirements of pediatric drug development for adult oncology products if directed at a molecular target also relevant to the growth or progression of pediatric cancer [40]. This contributes to a similar mandate to require pediatric research by EMA and FDA for products in the field of oncology. Follow-up research would be needed in light of these recent changes.

## Conclusion

This study found no significant differences in the available pediatric use information in EU SmPCs and USPIs for novel products approved in the period 2010–2018 by both FDA and EMA. Although the EU pediatric regulation gives a broader mandate for requiring pediatric drug development, this is not reflected in the prescription information approximately five years after authorization by the two regulatory authorities.

## Supporting information

**S1 Fig. Distribution of approval lag time between the EU and the US.**
(DOCX)

**S1 Table. Descriptive overview of therapeutic areas of products in total and with a pediatric indication in the EU and the US ($n$ = 217).**
(DOCX)

**S2 Table. Level of guidance for pediatric use for indications in SmPC and USPI for children ($n$ = 348).**
(DOCX)

**S3 Table. Level of guidance for pediatric use for indications in SmPC and USPI for toddlers and infants ($n$ = 348).**
(DOCX)

**S4 Table. Level of guidance for pediatric use for indications in SmPC and USPI for term newborn ($n$ = 348).**
(DOCX)

**S5 Table. Level of guidance for pediatric use per age group for indications without an orphan drug designation and/or pediatric indications outside of the adult indication in the EU and the US.** Percentages calculated from the total number of indications without an orphan drug designation and/or pediatric indications outside of the adult indication ($n$ = 186).
(DOCX)

**S6 Table. Concordance of the level of guidance for pediatric use in the SmPC and the USPI for adolescents for indications without an orphan drug designation and/or pediatric indications outside of the adult indications in the EU and the US ($n$ = 186).**
(DOCX)

**S1 Appendix. CIRS definition of New Active Substances (NAS).**
(DOCX)

**S1 Data. Data for the products included in the study ($n = 217$).**
(CSV)

**S2 Data. Data for the indications included in the study ($n = 348$).**
(CSV)

## Acknowledgments

We thank CIRS for providing us with a digital list of all new active substances approved by the EMA and the FDA. We also thank Anne Vinther Morant for providing insightful knowledge.

MedDRA® trademark is registered by IFPMA on behalf of ICH.

## Author Contributions

**Conceptualization:** Helle Christiansen, Marie L. De Bruin, Sven Frokjaer, Christine E. Hallgreen.

**Data curation:** Helle Christiansen.

**Formal analysis:** Helle Christiansen.

**Funding acquisition:** Marie L. De Bruin, Christine E. Hallgreen.

**Investigation:** Helle Christiansen, Christine E. Hallgreen.

**Methodology:** Helle Christiansen, Marie L. De Bruin, Sven Frokjaer, Christine E. Hallgreen.

**Project administration:** Helle Christiansen.

**Resources:** Helle Christiansen.

**Software:** Helle Christiansen.

**Supervision:** Marie L. De Bruin, Sven Frokjaer, Christine E. Hallgreen.

**Validation:** Marie L. De Bruin, Christine E. Hallgreen.

**Visualization:** Helle Christiansen, Christine E. Hallgreen.

**Writing – original draft:** Helle Christiansen, Christine E. Hallgreen.

**Writing – review & editing:** Marie L. De Bruin, Sven Frokjaer, Christine E. Hallgreen.

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
