## [Decision Letter · Decision Letter 0]

22 Nov 2021

PONE-D-21-08607Guidance for pediatric use in prescription information of novel therapeutics in the EU and the USPLOS ONE

Dear Dr. Christiansen,

Thank you for submitting your manuscript to PLOS ONE. After careful consideration, we feel that it has merit but does not fully meet PLOS ONE’s publication criteria as it currently stands. Therefore, we invite you to submit a revised version of the manuscript that addresses the points raised during the review process.

Please revise the manuscript in strict adherence with the Reviewer's comments.

We look forward to receiving your revised manuscript.

Kind regards,

Carlo Torti

Academic Editor

PLOS ONE

Journal Requirements:

5. Please upload a new copy of Figures 1 and 2 as the detail is not clear. Please follow the link for more information: " ext-link-type="uri" xlink:type="simple">https://blogs.plos.org/plos/2019/06/looking-good-tips-for-creating-your-plos-figures-graphics/"
" ext-link-type="uri" xlink:type="simple">https://blogs.plos.org/plos/2019/06/looking-good-tips-for-creating-your-plos-figures-graphics/"

Reviewers' comments:

Reviewer's Responses to Questions

**Comments to the Author**

1. Is the manuscript technically sound, and do the data support the conclusions?

Reviewer #1: Yes

Reviewer #2: Yes

2. Has the statistical analysis been performed appropriately and rigorously? 

Reviewer #1: Yes

Reviewer #2: Yes

3. Have the authors made all data underlying the findings in their manuscript fully available?

Reviewer #1: Yes

Reviewer #2: No

4. Is the manuscript presented in an intelligible fashion and written in standard English?

Reviewer #1: Yes

Reviewer #2: Yes

5. Review Comments to the Author

Reviewer #1: Dear Authors,

I have read your manuscript and I think that t is very well written, and references are well reported. The limitations of the study have been also described.

Therefore I have not comments regarding it

Reviewer #2: The comparison of the pediatric information provided in drug labels across the EU and the US covered in this paper is an incredibly valuable contribution given differences in pediatric regulations and incentives across these two regions. The findings of the paper are fascinating – it seems reasonable to begin with a prior that because of the broader scope of drugs and conditions covered by EMA regulations relative to FDA regulations, more information and more guidance would be provided in the EU. However, the authors find differences in information provided across the two regions, with an equal share of indications having more guidance in one region relative to the other. I would also like to applaud the authors for what must have been an extensive and careful data collection exercise. I have listed below a few major suggestions that I believe will strengthen the conclusions of the paper and a few more minor comments after that (mostly stylistic comments and suggestions related to clarity).

Major comments

• I find the three possible levels at which statistics are reported in this paper slightly disorienting, especially since the authors discuss in the data section that they map indications to disease/conditions, which makes it seem that the level of analysis is disease/condition. This is also the level of analysis that the authors state in their analysis section, and yet we only see statistics for drugs and indications. It would be helpful to use consistent terminology across the paper and in the results. Even when reporting both statistics for drugs and indications (or diseases/conditions), the emphasis should be on indications (or diseases/conditions) since that appears to be the main level of analysis.

• While I broadly agree with the tiered categories used for pediatric guidance, I also recommend adding a category on dosage or pharmacokinetic/pharmacodynamic studies. Many pediatric studies in the US may not focus on determining safety/efficacy but rather dosage or pharmacokinetics and pharmacodynamics, making it an important dimension of information to consider for comparing drug labels across the two regions.

• It’s not clear to me that the mandate covers more drugs/indications in the EU relative to the US and that we should expect differences in guidance across the two regions because of the BPCA. The authors do point out that the BPCA in the U.S. is designed so that studies can be requested for orphan drugs and other indications that are exempt under PREA. It is worth investigating the BPCA role further.

o First, to assess differences in guidance resulting from factors outside of the scope of PREA vs. PIP, please report the level of discrepancies and where discrepancies lie when excluding orphan drugs and pediatric indications outside of the adult indication. This would directly inform how much of the observed discrepancies are due to regulatory agency interpretation or approach, level of information provided, etc. rather than differences in where studies are requested across the two regions.

o How many of the drugs with more guidance in US were covered under PREA but have more information because of BPCA? The authors can add information on issued written requests under BPCA (which are publicly reported by the FDA) and how much of the observed discrepancies between EU and the US are reduced due to the BPCA.

• Are there any systematic differences in which types of drugs have more info in EU vs. which drugs have more info in the US? I would move Table 2 to appendix and add a table summarizing drug categories, disease categories, or sponsors where there seem to be differences.

Minor comments

• The quality of images for Figures 1 and 2 is low, making the figures illegible. It would be great if the authors could provide better quality images or figures.

• Please list in the data section the ages used for each pediatric group.

• Within each pediatric age group, I assume that the authors considered a given level of guidance as present in that age group if guidance was provided for any age within that group. For example, if guidance was provided for ages 15-17, then this would count as having adolescent use guidance, even if only providing guidance for a partial age range of the full adolescent age range. If I have understood this methodology correctly, it would be great to have this information included in the data section.

• A stylistic suggestion – in the abstract, a “broader mandate” sounds ambiguous because it is not obvious what the breadth refers to. “Mandate of a broader scope” or a version of this wording to indicate that the EMA can require pediatric assessments in more cases than the FDA might be clearer.

• In the abstract, it takes a few reads to understand that the focus is on indications listed as of March 2020 for products approved 2010-2018. Please make the wording clearer.

• Throughout the paper, “higher level of guidance” is slightly odd to use because guidance is either provided or not, but the level of information in the guidance can differ. I would recommend using terminology referring to the information levels rather than guidance levels (e.g., “more information” or “higher level of information”).

• Throughout the paper, the authors should use “an equal share” instead of “an equal distribution” for simplicity and clarity.

• Line 77 typo: Cross jurisdiction investigations is important � Cross jurisdiction investigations are important.

• I would recommend either moving to appendix or fully excluding the analysis, comparison, and discussion of withdrawn products since it detracts from the main point of the paper (unless these are products who were withdrawn due to issues in the pediatric population, in which case please discuss these).

• To tie the results in directly to the motivation on different regulations across the two regulations, it would be great to mention in the abstract that only 21% of discrepancies in pediatric information can be explained by differences in regulations.

6. PLOS authors have the option to publish the peer review history of their article (what does this mean?). If published, this will include your full peer review and any attached files.

Reviewer #1: No

Reviewer #2: No

---

## [Author Response · Author response to Decision Letter 0]

14 Feb 2022

Response to Reviewer

Journal Requirements:

Answer: The manuscript has been revised to meet PLOS ONE’s style requirements. In addition, we have also revised the language. We have changed the use of ´therapeutics´ to ´medicinal products´ and ´products´ to for consistency in the terms used. All updates to the manuscript have been made with track changes, including improvements based on reviewer comments.

Answer: We have submitted the manuscript in a .docx format according to guidelines and in our opinion, the comment is no longer applicable.

Answer: The grant provided by H. Lundbeck A/S does not have a grant number. H. Lundbeck A/S has contributed with financial means for 1 Ph.D. student in supporting the University of Copenhagen, Faculty of Health and Medicinal Sciences, in its ´Copenhagen Centre for Regulatory Science´ (CORS) initiative. The funding was given from H. Lundbeck A/S to CORS. The Ph.D. student, Helle Christiansen is employed by CORS and her research is purely devoted to the scientific aspects of the regulatory field. The research is not a company-specific product or directly company-related. H. Lundbeck A/S had no role in study design, data collection, and analysis, decision to publish, or preparation of the manuscript. We have added a comment in the funding information that no grand number exists.

Answer: All relevant data are in Supporting Information files. 

5. Please upload a new copy of Figures 1 and 2 as the detail is not clear. Please follow the link for more information: https://blogs.plos.org/plos/2019/06/looking-good-tips-for-creating-your-plos-figures-graphics/" https://blogs.plos.org/plos/2019/06/looking-good-tips-for-creating-your-plos-figures-graphics/"

Answer: Figures 1 and 2 have been uploaded to the Preflight Analysis and Conversion Engine (PACE) digital diagnostic tool to ensure that figures meet PLOS requirements. They are now in a TIFF format with correct dimensions and resolution. 

Reviewers’ comments:

Reviewer #1: Dear Authors,

I have read your manuscript and I think that t is very well written, and references are well reported. The limitations of the study have been also described.

Therefore I have not comments regarding it.

Reviewer #2: The comparison of the pediatric information provided in drug labels across the EU and the US covered in this paper is an incredibly valuable contribution given differences in pediatric regulations and incentives across these two regions. The findings of the paper are fascinating – it seems reasonable to begin with a prior that because of the broader scope of drugs and conditions covered by EMA regulations relative to FDA regulations, more information and more guidance would be provided in the EU. However, the authors find differences in information provided across the two regions, with an equal share of indications having more guidance in one region relative to the other. I would also like to applaud the authors for what must have been an extensive and careful data collection exercise. I have listed below a few major suggestions that I believe will strengthen the conclusions of the paper and a few more minor comments after that (mostly stylistic comments and suggestions related to clarity).

Major comments

• I find the three possible levels at which statistics are reported in this paper slightly disorienting, especially since the authors discuss in the data section that they map indications to disease/conditions, which makes it seem that the level of analysis is disease/condition. This is also the level of analysis that the authors state in their analysis section, and yet we only see statistics for drugs and indications. It would be helpful to use consistent terminology across the paper and in the results. Even when reporting both statistics for drugs and indications (or diseases/conditions), the emphasis should be on indications (or diseases/conditions) since that appears to be the main level of analysis.

Answer: The methods section has been strengthened to ensure consistent terminology. We investigated guidance for pediatric use for all indications for products approved in both regions during the study period. The study unit has been clarified to be “product-indication” which is simplified to “indication” throughout the manuscript. The indication is the only level used for analysis, which has been clarified under “Analysis” in the method section. 

The condition or disease (depending on the level mentioned in the SmPC or USPI) were only used to identify the product-indications to compare between the regions. 

• While I broadly agree with the tiered categories used for pediatric guidance, I also recommend adding a category on dosage or pharmacokinetic/pharmacodynamic studies. Many pediatric studies in the US may not focus on determining safety/efficacy but rather dosage or pharmacokinetics and pharmacodynamics, making it an important dimension of information to consider for comparing drug labels across the two regions.

Answer: We acknowledge the useful information in having a category on dosage or pharmacokinetic/pharmacodynamic studies. We did investigate the possibilities of separating human data available according to real world data or pharmacokinetic/pharmacodynamic studies. However, from the SmPCs and the USPIs, it is not always apparent and providing this information would be too much of a guess resulting in analyses that lack robustness.

We address this point in the discussion with a suggestion for future study: “Further research is needed to assess the extent to which pediatric development has resulted in new posology guidance for the pediatric and possible new age-appropriate formulations”.

• It’s not clear to me that the mandate covers more drugs/indications in the EU relative to the US and that we should expect differences in guidance across the two regions because of the BPCA. The authors do point out that the BPCA in the U.S. is designed so that studies can be requested for orphan drugs and other indications that are exempt under PREA. It is worth investigating the BPCA role further.

o First, to assess differences in guidance resulting from factors outside of the scope of PREA vs. PIP, please report the level of discrepancies and where discrepancies lie when excluding orphan drugs and pediatric indications outside of the adult indication. This would directly inform how much of the observed discrepancies are due to regulatory agency interpretation or approach, level of information provided, etc. rather than differences in where studies are requested across the two regions.

Answer: Thank you for this highly relevant request. We have discussed this matter several times during the preparation of the study. If we exclude indications with an orphan drug designation or pediatric indication approved outside the adult indication, we exclude those indications where we expect the largest difference in guidance for pediatric use based on the differences in the mandatory pediatric framework. In our study, we try to report all the differences in the guidance for pediatric use, and map which differences could be caused by differences in the framework of mandatory pediatric legislations.

Initially, we did make the analysis suggested by the reviewer, however, we decided not to put it into the supplementary tables since this analysis showed the same pattern as the analysis on the entire sample (all indications). As part of this revision, we have expanded the supplementary tables to include tables showing the level of discrepancies and where discrepancies lie when excluding orphan drugs and pediatric indications outside of the adult indication. 

o How many of the drugs with more guidance in US were covered under PREA but have more information because of BPCA? The authors can add information on issued written requests under BPCA (which are publicly reported by the FDA) and how much of the observed discrepancies between EU and the US are reduced due to the BPCA.

Answer: Thank you for this valid point. We did investigate this; however, an issued written request does not ensure that the studies in the written request are being conducted. Pediatric drug development requested through the BPCA will only be conducted if the companies take on the development and this information is not available before a pediatric exclusivity has been granted. Further, the content of a written request is also not available before a pediatric exclusivity has been granted. Therefore, we cannot collect the information on indication level, but only on product-level. See the example with Lurasidone below. 

When removing the indications with an orphan drug designation or indications outside of the adult indication, 186 indications are left. Of these, a total of 30 indications has a difference in the guidance for use for one or more subgroups of the pediatric population (see Supplementary Tables 5 and 6), distributed with 16 indications with a “use” or “do not use” in the US as compared to “human data available” or “no guidance provided” in the EU. 

A WR has been issued for 8 active substances covering 9 indications with “more” guidance for use in the US, however, only 4 active substances had been granted a pediatric exclusivity. For the active substance Lurasidone, the written request made public upon granting the pediatric exclusivity reveals that the WR only concerns schizophrenia even though Lurasidone had been granted approval for both schizophrenia and bipolar I disorder with major depressive episodes. Therefore, until a WR has been made public, we can only guess which indications that are covered in the WR, making it unreliable for reporting. 

• Are there any systematic differences in which types of drugs have more info in EU vs. which drugs have more info in the US? I would move Table 2 to appendix and add a table summarizing drug categories, disease categories, or sponsors where there seem to be differences.

Answer: We did not see any systematic differences in which types of drugs have more guidance for pediatric use in the EU and vice versa.

Table 2 has been changed to include only the therapeutic areas where we saw a difference between the granted pediatric indications. The full table is moved to supplementary files.

Minor comments

• The quality of images for Figures 1 and 2 is low, making the figures illegible. It would be great if the authors could provide better quality images or figures.

Answer: We have provided new figures with better quality. We are very sorry that we did not provide this in the first submission.

• Please list in the data section the ages used for each pediatric group.

Answer: Ages have been listed in the method section. The new wording is as follows: “The type of pediatric information was captured for each age group of the pediatric population (adolescents (12-18 years), children (2-11 years), infants and toddlers (28 days to 23 months), and term newborn (0-27 days)) as defined by International Conference on Harmonization (ICH) Topic E 11, 2001 [27].”

• Within each pediatric age group, I assume that the authors considered a given level of guidance as present in that age group if guidance was provided for any age within that group. For example, if guidance was provided for ages 15-17, then this would count as having adolescent use guidance, even if only providing guidance for a partial age range of the full adolescent age range. If I have understood this methodology correctly, it would be great to have this information included in the data section.

Answer: It is correctly understood, and the methods section has been updated. The following sentence was added: “The level of guidance was considered to cover a certain age group if guidance was provided for any age within that group.”

• A stylistic suggestion – in the abstract, a “broader mandate” sounds ambiguous because it is not obvious what the breadth refers to. “Mandate of a broader scope” or a version of this wording to indicate that the EMA can require pediatric assessments in more cases than the FDA might be clearer.

Answer: Thank you for the suggestion. We have changed the wording into the following: “Despite many similarities between these frameworks, the EU Pediatric Regulation more often provides regulators with a mandate to require pediatric drug development for novel therapeutics compared to US regulators.” 

• In the abstract, it takes a few reads to understand that the focus is on indications listed as of March 2020 for products approved 2010-2018. Please make the wording clearer.

Answer: Thank you for the feedback. We tried to make the wording clearer and hopefully, it will now be understandable after the first read. The wording has been changed into: “For all indications granted as of March 2020 for novel therapeutics approved in both regions between 2010 and 2018, we compared the guidance for pediatric use in the EU SmPC and the USPI.”

• Throughout the paper, “higher level of guidance” is slightly odd to use because guidance is either provided or not, but the level of information in the guidance can differ. I would recommend using terminology referring to the information levels rather than guidance levels (e.g., “more information” or “higher level of information”).

Answer: Thank you for pointing this out. We have changed the phrasing to “higher level of information”. 

• Throughout the paper, the authors should use “an equal share” instead of “an equal distribution” for simplicity and clarity.

Answer: We have changed “an equal distribution” to “an equal share” throughout the manuscript. 

• Line 77 typo: Cross jurisdiction investigations is important � Cross jurisdiction investigations are important.

Answer: The typo has been corrected according to the reviewer’s request.

• I would recommend either moving to appendix or fully excluding the analysis, comparison, and discussion of withdrawn products since it detracts from the main point of the paper (unless these are products who were withdrawn due to issues in the pediatric population, in which case please discuss these).

Answer: Thank you for highlighting this. Products were not withdrawn due to issues in the pediatric population and all 14 products with a withdrawal in one or both regions have the similar guidance for pediatric use. Therefore, having them in or out of the analysis makes no difference to our findings. We discussed this before the data collection, when conducting the analysis and drafting the manuscript. We kept them in because it was important for us to investigate the total number of differences between the regions. 

To removed focus on the withdrawn products, we have deleted “Withdrawn” from Table 1 and we have removed the sentence “At the end of follow up (March 2020), 13 products had been withdrawn in the EU, and six in the US. Of these products, five were withdrawn in both regions” has been removed from the manuscript. 

• To tie the results in directly to the motivation on different regulations across the two regulations, it would be great to mention in the abstract that only 21% of discrepancies in pediatric information can be explained by differences in regulations.

Answer: Thank you for making this valid point. We have made changes in the abstract to include the above-mentioned information noticed by the reviewer. The sentence added to the abstract is: “The discrepancies in pediatric information could possibly be explained by differences in regulations for 21% (13/61) of the indications.”

---

## [Decision Letter · Decision Letter 1]

21 Mar 2022

Guidance for pediatric use in prescription information for novel medicinal products in the EU and the US

PONE-D-21-08607R1

Dear Dr. Christiansen,

We’re pleased to inform you that your manuscript has been judged scientifically suitable for publication and will be formally accepted for publication once it meets all outstanding technical requirements.

Kind regards,

Carlo Torti

Academic Editor

PLOS ONE

Additional Editor Comments (optional):

Reviewers' comments:

Reviewer's Responses to Questions

**Comments to the Author**

1. If the authors have adequately addressed your comments raised in a previous round of review and you feel that this manuscript is now acceptable for publication, you may indicate that here to bypass the “Comments to the Author” section, enter your conflict of interest statement in the “Confidential to Editor” section, and submit your "Accept" recommendation.

Reviewer #2: All comments have been addressed

2. Is the manuscript technically sound, and do the data support the conclusions?

Reviewer #2: (No Response)

3. Has the statistical analysis been performed appropriately and rigorously? 

Reviewer #2: (No Response)

4. Have the authors made all data underlying the findings in their manuscript fully available?

Reviewer #2: (No Response)

5. Is the manuscript presented in an intelligible fashion and written in standard English?

Reviewer #2: (No Response)

6. Review Comments to the Author

Reviewer #2: (No Response)

7. PLOS authors have the option to publish the peer review history of their article (what does this mean?). If published, this will include your full peer review and any attached files.

Reviewer #2: No

---

## [Editor Report · Acceptance letter]

25 Mar 2022

PONE-D-21-08607R1 

Guidance for pediatric use in prescription information for novel medicinal products in the EU and the US 

Dear Dr. Christiansen:

I'm pleased to inform you that your manuscript has been deemed suitable for publication in PLOS ONE. Congratulations! Your manuscript is now with our production department. 

Kind regards, 

on behalf of

Dr. Carlo Torti 

Academic Editor

PLOS ONE